# Health Care Responsibility and Compassion-Visiting the Housebound Patient Severely Affected by ME/CFS

**DOI:** 10.3390/healthcare8030197

**Published:** 2020-07-04

**Authors:** Caroline Kingdon, Dionysius Giotas, Luis Nacul, Eliana Lacerda

**Affiliations:** Department of Clinical Research, London School of Hygiene & Tropical Medicine, Faculty of Infectious and Tropical Diseases, London WC1E 7HT, UK; dio.giotas@lshtm.ac.uk (D.G.); luis.nacul@lshtm.ac.uk (L.N.); Eliana.Lacerda@lshtm.ac.uk (E.L.)

**Keywords:** ME/CFS, severe ME/CFS, validation, engagement, health encounters, housebound, bedbound

## Abstract

Many people with severe Myalgic Encephalopathy/Chronic Fatigue Syndrome (ME/CFS) commonly receive no care from healthcare professionals, while some have become distanced from all statutory medical services. Paradoxically, it is often the most seriously ill and needy who are the most neglected by those responsible for their healthcare. Reasons for this include tensions around the complexity of making an accurate diagnosis in the absence of a biomarker, the bitter debate about the effectiveness of the few available treatments, and the very real stigma associated with the diagnosis. Illness severity often precludes attendance at healthcare facilities, and if an individual is well enough to be able to attend an appointment, the presentation will not be typical; by definition, patients who are severely affected are home-bound and often confined to bed. We argue that a holistic model, such as ‘‘Compassion in Practice’’, can help with planning appointments and caring for people severely affected by ME/CFS. We show how this can be used to frame meaningful interactions between the healthcare practitioners (HCPs) and the homebound patient.

## 1. Introduction

### 1.1. ME/CFS and the Needs of the Severely Affected Individual

ME/CFS is a complex and multifactorial disease; the World Health Organisation has described ME/CFS as a neurological disorder in the International Classification of Functioning, Disability and Health since 1969 [1]. It is also highly stigmatised [2], and there is currently neither a validated biomarker nor an effective treatment. Disease manifestation can vary between affected individuals, and some routine clinical examinations and blood tests may not evidence abnormalities [3]. However, a careful clinical examination, which should be part of a clinical assessment, may show subtle abnormalities [4]. People with severe ME/CFS, estimated as 25% of those affected [5], have been defined as being almost exclusively housebound and unable to attend healthcare consultations, and often bedbound all or some of the time [4]. Multiple studies worldwide [6] have uncovered abnormalities in the central and autonomic nervous system, as well as changes to the metabolic and immunological systems. Severe or striking fatigue, which is often disabling, must have been present for at least six consecutive months (in adults) in order to meet currently accepted diagnostic guidelines [7,8,9]. While ME/CFS can affect multiple systems, many people with mild/moderate ME/CFS continue to function “deceptively normally” by carefully planning their activities and periods of rest. This may involve, for example, managing to hold down a job but not taking part in any social activities in the evenings and at weekends so that they can rest, significantly impacting friendships and quality of life. However, other patients are severely functionally impaired, and physical or mental exertion beyond a personal threshold can exacerbate symptoms; this is commonly known as post-exertional malaise (PEM). PEM can be debilitating and increases the individual’s dependency on others for hours, days, weeks or even months. The vocabulary to describe the experience of people with severe ME/CFS is inadequate, and words such as malaise or fatigue diminish and deny the grim reality of the disease.

ME/CFS has a prevalence of between 0.2 and 0.4%; recent estimates of those affected in the UK give a figure of between 150,000 and 250,000 people [10]. Two to four times as many women as men are affected, which is similar to multiple sclerosis (MS) and some other autoimmune diseases, where major fatigue is also a symptom. A GP practice of 10,000 patients could have up to 40 people with ME/CFS (PWME), with 25% severely affected [5]. PWME have been found to be more functionally impaired than people with cancer or other chronic diseases [11]. ME/CFS is potentially life-shortening, although registered deaths from ME/CFS are rare; between 2001 and 2016, 88 deaths in England and Wales were partly or fully attributable to ME/CFS [12]. This is a disease of low prestige, and PWME often experience damaging negative encounters with GPs, nurses, and other healthcare practitioners (HCPs) [13]; disrespectful treatment and trivialising of legitimate symptoms may lead to loss of agency and alienation from all medical services. PWME are often made to feel, in the words of one parent carer, “*that they are second-class citizens, malingerers or at least somehow responsible for their own misfortune*” [14]. In the face of such discrimination, we strongly believe that HCPs are duty-bound to ensure that the person with severe ME/CFS receives the person-centred care she/he needs, despite the fact that, as is understood currently, little can be done to change the pattern of the illness.

As part of the CureME research team at the London School of Hygiene and Tropical Medicine, we have visited a cohort of nearly 100 patients housebound by ME/CFS on multiple occasions, in order to enrol them in the UK ME/CFS Biobank (UKMEB) [15]. After considering our experiences and insights from these unique encounters in a methodical way, we argue that a holistic model, such as ‘‘Compassion in Practice’’ [16], can help with planning and caring for people severely affected by ME/CFS. We show how this can be used to frame meaningful interactions between the HCPs and the homebound patient.

To substantiate this discussion, we present in Table 1 an analysis of the frequency and severity of symptoms reported by data available from an early subsample of our cohort with severe presentation of ME/CFS (*n* = 57), whose gender breakdown reflects the accepted 3:1, F:M ratio [10]. These are accompanied by revealing quotes from those visited by the CureME team:

### 1.2. Responsibility and Compassion

Responsibility and compassion should guide all interactions between HCPs and patients and are critical to the therapeutic relationship; they are particularly important when HCPs interact with people with severe ME/CFS. The following approaches exemplify the ways that attributes, including empathy, respect and dignity, can be evident in many practical tasks when medical encounters are carefully planned and executed.

#### 1.2.1. Arranging a Visit

Most appointments, either face-to-face or virtual, are ideally scheduled for after midday as many people with severe ME/CFS need to rest until then due to erratic sleep patterns (a). Connecting with those most severely affected may require some persistence, and the practitioner should always ensure that his/her tone of voice (k) is appropriate for someone seriously ill, and that questions are simple, requiring short answers. People with ME/CFS often make an enormous effort to prepare for the HCP visit, and the post visit malaise and exhaustion (c, d) is never witnessed by the visiting HCP but only by those who remain with the patient. The PEM caused by such efforts may continue for hours, days or even weeks.

Looking after people with severe ME/CFS at home may require ongoing commitment from the HCP over many years, as improvement in the condition is uncommon; the HCP may need to schedule short visits at regular intervals, aiming to address one or two problems at each visit commensurate with the health of the individual.

#### 1.2.2. Preparing for a Visit

When planning the visit to the person with severe ME/CFS, the HCP should avoid smoking or the use of perfumes or fragranced lotions, as sensitivities to chemicals or odours (l) are common (4, 5); patients may even request that clothes are not freshly laundered because of the effect of fragranced detergents. HCPs with symptoms of viral infections should avoid visits, as there is evidence that the altered immunological status in ME/CFS makes people more susceptible to infection [17]. On entering the home of any person with severe ME/CFS, the HCP should offer to remove shoes and be seen to undertake thorough hand hygiene.

Visiting someone with severe ME/CFS necessitates use of the HCP’s experience and expertise to priorities the patient’s agenda according to need, to make the visit profitable for both parties. In the current absence of effective treatments for ME/CFS, the person with severe ME/CFS is often very aware of which strategies help or hinder their ability to function. ME/CFS charities (see Table 2) are a rich resource, and the well-informed HCP should know how to access websites, printing information in advance to hand out if appropriate and helpful.

#### 1.2.3. The Home Visit

The visiting practitioner should assess the disease severity and what that means for the patient using observational skills, without judgment or agenda, to better understand the health and social needs of the individual. The observant HCP will seek clues about who the individual was before the onset of disease; patients have often had to redefine their sense of self out of necessity [18]. A common consequence of the disease is physical, emotional and intellectual isolation; family and friends are often estranged, partnerships and marriages flounder, and studies are curtailed and jobs are lost, with ensuing loss of income [19].

HCPs without appropriate expertise may be reluctant to engage with PWME leading to inequity in treatment. PWME are often better informed about the disease than the visiting practitioner. Lack of support from established medical services force many to draw on e-resources, including support groups and online forums. In the absence of validation and support from the medical community, some PWME self-manage the illness, gathering information to legitimise the illness [20] and in an attempt to find sources of help. Taking a proactive stance with their own health differentiates PWME from people with clinical depression, and HCPs need to understand that it is the disease that inhibits activity (b, c) rather than lack of motivation. This can challenge the practitioner used to steering the therapeutic relationship who may need to acknowledge their limitations. People with severe ME/CFS are as susceptible to other diseases as the general population, and may be at higher risk of some diseases [21]. It is therefore important that PWSME are thoroughly medically reviewed on a regular basis, as well as when there is deterioration in their health status, or when presenting signs or symptoms that could be due to new comorbidities, which could be treated.

“Brain fog” (h) or cognitive dysfunction is common in PWME; time and patience are required to help the patient to make sense of his/her illness. It is often difficult and time-consuming to take a coherent medical history, not only because information processing and concentration skills (f, g, h, i, j) may be diminished by the disease but also because this is a disease with no clear natural history, in which symptoms can vary both from person to person and from day to day, over the course of the disease. Listening and hearing imply trust, which is essential to any therapeutic relationship. The individual with severe ME/CFS may be familiar with disbelief and skepticism, and therapeutic progress often involves belief in the patient’s symptoms and feelings.

Some people with severe ME/CFS may lie in a darkened room with earplugs in and may exhibit increased sensitivity to light, sound and touch (k, m, n); struggle to share information (f, h); or be so wary of PEM that they are fearful of articulating the things that they want or need to share (c). Others may seem animated and engaged, making it difficult for the practitioner to recognize the extent of their illness. Practitioners should use tone of voice (k) and pressure of touch appropriate to the individual’s sensitivities.

Apart from frailty, physical examination generally reveals little, and blood work, important to exclude other diagnoses, is often normal. Some severely affected individuals may appear undernourished; this often follows food intolerance and allergies (o) and is rarely a manifestation of anorexia nervosa [22]. People with severe ME/CFS may only be able to stand (e) for a few moments, if at all, and signs of weakness may be obvious. Handgrip measures are markedly reduced in the severely affected and rapidly decline when repeated [23].

The HCP should encourage balance between activity and rest, including during the encounter, to minimize any subsequent PEM and pain (c, d). There are times when the patient may risk PEM to accomplish something, and the HCP can guide the planning of such activities to minimize the risk of PEM. Such activities may, for the most severely affected, simply include producing a urine sample or raising an arm to have a BP cuff put on.

People with severe ME/CFS are disabled by their symptoms (b), and this may be an opportunity for the HCP to offer information about and access to disability aids and allowances. In the UK, benefits include Attendance Allowance and Personal Independence Payments. The ME/CFS charities are a rich resource for further information (Table 2).

Living with severe ME/CFS requires enormous courage. The inevitable reduction or loss of a societal role means that people with ME/CFS are unable fully to engage with family and friends. Not only do people with severe ME/CFS have to come to terms with a long-term, disabling illness, but they may also have to cope with doubts about its authenticity. Reactive depression and anxiety are common for people with severe ME/CFS as in any long-term, disabling illness, and suicide is a possible outcome in this patient group [24], so the HCP should always be sensitive to signs of deteriorating mental health and treat it appropriately.

The therapies and alternative treatments that people with severe ME/CFS sometimes embrace can seem unscientific to the HCP, but are sometimes turned to out of desperation; unless they could be detrimental to the individual’s health or involve unreasonable expense, the HCP should endeavor to support the patient in his/her chosen path. When patients face financial difficulties, the HCP must continue to support the severely affected individual to help them to manage that possible additional burden.

## 2. Conclusions

We believe that compassion is central to the care of people with ME/CFS. Despite the current absence of curative treatments for people with severe ME/CFS, the HCP has a responsibility to provide care through a relationship based on empathy, respect and dignity. By supporting the individual with compassion and competence and acknowledging and learning from the patient’s experience, the encounter with the housebound patient can be both effective and worthwhile. The first step in the therapeutic relationship is to believe and trust the individual: to articulate that you, the practitioner, hear what your patient is saying and recognise that their experience is legitimate.

This severe, complex multisystem disease has long been misjudged by the healthcare profession. Educating practitioners about the needs of those most severely affected by ME/CFS will help drive the step change in understanding and belief, compassion and empathy required to care for all patients with ME/CFS.

## Figures and Tables

**Table 1 healthcare-08-00197-t001:** Common symptoms affecting people with severe Myalgic Encephalopathy/Chronic Fatigue Syndrome (ME/CFS) seen by CureME.

Symptom		PWSME * with Symptom	Symptom Reported as Severe (%)
		(*n* = 57; 44 F, 13M)	
a	Unrefreshing sleep	100	70
“When I waken, often in the very late morning, I feel no more rested than I did at the end of the day before; everything is an effort no matter how long the time spent asleep.” (F aged 20–29)			
b	Disabling fatigue	100	n/a
	Sleep problems	83	65
“I can only describe a good day as trying to walk through water; on a bad day, the water becomes treacle.” (M aged 40–49)			
c	Exercise intolerance (PEM)	100	90
“If my daughter cleans her teeth herself, that is it for a week. She can’t do anything more.” (Parent describing F aged 40–49)			
d	Pain after exertion/activity	84	70
	Muscle pain	96	50
“I would love to know what one, single day without pain was like; I have quite forgotten. This pain is relentless.” (F aged 18–30)			
e	Intolerance to standing	81	50
“Standing is a nightmare–my heart pounds, my legs shake within minutes, I become dizzy and feel that I will faint.” (M aged 50–59)			
f	Concentration problems	96	43
“Reading is like trying to listen to the radio when there is so much interference that you can only hear the odd word…” (F aged 40–49)			
g	Difficulty in finding words	96	32
	Difficulty in making decisions	75	35
“When someone visits, I have to choose. I can either eat a biscuit and have a cup of tea or listen and respond. I cannot do both.” (F aged 30–39)			
h	Brain fog/cognitive dysfunction	95	45
	Slow thinking	88	32
“Sometimes I grasp at words, feeling like a toddler learning to speak all over again. I was studying for my PhD when I got ill.” (F aged 30–39)			
i	Short-term memory problems	95	33
“I am so sorry. I know you just asked me to do something, but I can’t remember what.” (F aged 40–49)			
j	Difficulty in understanding	86	31
	Difficulty in retaining information	89	35
“I have to write that down straight away or I forget what you said. And I forget I need to write it down!” (M aged 30–39)			
k	Unusual sensitivity to light and/or noise	91	45
“Would you mind just shining a light on my arm to take blood? Away from my eyes? I just can’t tolerate any light.” (M aged 20–29)			
“The whole family suffers because I can’t bear music playing or the chatter of voices–and yet I long to enjoy those things again.” (F aged 20–29)			
l	New sensitivities to food, medication, chemicals or odours	88	43
“Please don’t wear freshly washed clothes as the smell of detergent or softener can worsen her symptoms.” (Parent of F aged 50–59)			
m	Intolerance to heat and cold	81	57
“I sometimes think this must be what the menopause is like for women. I vacillate between hot sweats and feeling icy cold within minutes–for no apparent reason” (M aged 40–49)			
n	Allergies/hypersensitivities	80	63
“I have had to rearrange the appointment as she is still unable to leave the house due to the severe reaction to clothes and anything that touches her skin-you can understand that is a big, big problem.” (Parent of F aged 18–30)			
o	Gastrointestinal symptoms	79	40
	Sickness and nausea	79	16
“I became unable to tolerate different types of food-even the smell of food being cooked. Even water made me feel sick” (F aged 30–39)			

* PWSME—people with severe Myalgic Encephalomyelitis/Chronic Fatigue Syndrome. The letter to the left of a particular symptom is used to reference the symptom in the text following the table.

**Table 2 healthcare-08-00197-t002:** Examples of UK charities providing guidance for healthcare practitioners and/or patient resources for ME/CFS.

ME/CFS Charity	Webpage
ME Association	https://www.meassociation.org.uk/
Action for M.E.	https://www.actionforme.org.uk/
The ME Trust	https://www.metrust.org.uk/
25% M.E. Group-Supporting Those with Severe M.E.	https://25megroup.org

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
