# Peer review of "Health Care Responsibility and Compassion-Visiting the Housebound Patient Severely Affected by ME/CFS"

_healthcare, 2020, doi:10.3390/healthcare8030197_

Round 1

Reviewer 1 Report

17           Is “statutory” (regulated by law) the correct word? Established perhaps?

32           and some may appear, in most other respects, to be healthy. Vague statement. Be more specific. Patients can appear “healthy” at times, e.g., during remission, but it is generally agreed that this does not mean they are cured.

Many routine clinical tests?

41           for example, this may  involve holding down a job but not taking part in any social activities in the evenings and at weekends so that they can rest, all of which significantly impacts friendships and quality of life. Run on sentence.

Suggest: they can rest. All of which significantly impacts

44          However, other patients are severely functionally impaired, and physical or mental exertion beyond a personal threshold can exacerbate symptoms, which may be debilitating, increasing their dependency on others, for hours, days, weeks or even months; this is commonly known as post exertional malaise (PEM). Run on sentence.

52          major fatigue is also a symptom. A GP practice of 10,000 patients could have up to 40 PWME, (First use of acronym  should include full term)

53          25% severely affected [5]. PWME have been found to be more functionally impaired than people with cancer or other chronic diseases [12] and ME/CFS is potentially life-shortening, although registered deaths from ME/CFS are rare; between 2001 and 2016, 88 deaths in England and Wales were partly or fully attributable to ME/CFS [13]. Run on sentence.

71          To substantiate this discussion, we present in Table 1 an analysis of the frequency and severity of symptoms reported by data available from an early subsample of our cohort with severe presentation of ME/CFS (n=57), which are accompanied by revealing quotations (quotes?) from those visited by the CureME team:

Please provide gender breakdown for sample (n=57) and symptom responses. Discuss gender implications where appropriate.

89          visit and the post-visit malaise and exhaustion (c,d) is witnessed only by those who stay (explain)

118        HCPs may be reluctant to engage with PWME because of lack of expertise leading to inequity in treatment. PWME are often better informed about the disease than the visiting practitioner, because lack of support from established medical services force many to draw on e-resources, including support groups and online forums.

Suggest: HCPs who lack expertise may be reluctant to engage with PWME leading to inequity in treatment. PWME are often better informed about the disease than the visiting practitioner.  Lack of support from established medical services force many to draw on e-resources, including support groups and online forums.

128        and may be at higher risk of some diseases [22]. It is therefore important they are thoroughly medically reviewed when presenting signs or symptoms or (delete, or is there a missing phrase?) that may indicate alternative diseases and when there is deterioration in their health status.

151        PEM and pain after activity are cardinal symptoms of ME/CFS (c, d) and even in the encounter, the HCP should encourage balance between activity and rest.

Suggest: The HCP should encourage balance between activity and rest, including during the encounter.

156        People with severe ME/CFS are disabled by their symptoms (b), and this may be an opportunity for the HCP to offer information about and access to disability aids and allowances. In the UK, benefits include Attendance Allowance and Personal Independence Payments. (Can you reference sources for further information on disability aids and allowances?)

161  people with severe ME/CFS ( have to) must come to terms with a long-term, disabling illness, but they may

The paper does identify a legitimate problem in the care of PWME. However, I believe that a solution will require  institutional support for HCP. A paragraph on suggestions for improving HCP ME education and training that goes beyond merely asking them to hear what patients are saying would add to the paper.

Paper may be improved with further editing with attention given to sentence structure. Avoid run on and/or overlong sentences.

Reviewer 2 Report

  • Abstract: Severity of illness precludes attendance at health care facilities should be included. This is addressed satisfactorily in the manuscript text but could also be mentioned in the Abstract due to its importance in patient decision making and access to healthcare
  • Line 30. 'No biomarker' is not completely correct. The authors might include that deficits in natural killer (NK) cell function have been widely reported and may indicate pathology more generally and become a useful biomarker. These findings are important to note as pharmacotherapeutics can be applied in vitro to alter cell function and hence may improve illness trajectory when accepted in clinical practice subsequently.
  • Use of illness vs disease. Currently as fully described pathology is lacking in ME/CFS it may be prudent to use the word illness rather than disease.
  • PWME not spelt at first use. Patients/people with ME?

Author Response

Use of illness vs disease. Currently as fully described pathology is lacking in ME/CFS it may be prudent to use the word illness rather than disease.

Thank you for your comment. I appreciate that the use of disease seems clumsy at times in this paper, but I believe it is the correct term, particularly in the context of illness belief. May I respectfully refer you to a paper in the RCGP journal?

 https://www.ncbi.nlm.nih.gov/pmc/articles/PMC1972172/

 The terms 'disease' and 'illness' are used by medical  anthropologists to describe the different views of ill-health held by doctors and their patients.

Line 52          major fatigue is also a symptom. A GP practice of 10,000 patients could have up to 40 PWME, (First use of acronym  should include full term)

Thank you for pointing tout my use of an abbreviation with no explanation. My apologies for this sloppiness. (58)

Reviewer 3 Report

This paper is a valuable contribution to a subject area of major concern for those afflicted with this condition.

I would suggest rewriting the opening sentence of the Abstract.As it stands it seems to blame the patients for the lack of interaction with HCPs. Instead, it should be stressed that this failing is the responsibility of the HCPs themselves, as they paradoxicaly neglect some of their most seriously ill and needy patients

It is a pity the authors did not use their contact with their patient group to actually measure what proportion of them were suffering near total neglect by their HCPs. If this information was available it should be added to the paper at this stage

Author Response

 I would suggest rewriting the opening sentence of the Abstract.As it stands it seems to blame the patients for the lack of interaction with HCPs. Instead, it should be stressed that this failing is the responsibility of the HCPs themselves, as they paradoxicaly neglect some of their most seriously ill and needy patients

It is a pity the authors did not use their contact with their patient group to actually measure what proportion of them were suffering near total neglect by their HCPs. If this information was available it should be added to the paper at this stage.

The authors thank the reviewer for all the comments and suggestions, which are warmly received and most helpful.

Here is my revised opening paragraph, which I hope responds to your concerns: Many people with severe Myalgic Encephalopathy/Chronic Fatigue Syndrome (ME/CFS) commonly receive no care from healthcare professionals, while some have become distanced from all statutory medical services. Paradoxically, it is often the most seriously ill and needy who are the most neglected by those responsible for their healthcare. Reasons for this include tensions around the complexity of making an accurate diagnosis in the absence of a biomarker, the bitter debate about the effectiveness of the few available treatments, and the very real stigma associated with the diagnosis. Illness severity often precludes attendance at healthcare facilities, and if an individual is well enough to be able to attend an appointment, the presentation will not be typical; by definition, patients who are severely affected are home-bound and often confined to bed.

Regarding measuring lack of contact with GPs:

One requirement of taking part in the study was that participants had been given a clinical diagnosis of ME/CFS. They were also asked to provide the name and address of their general practitioner in order that the results of routine blood tests could be shared if they were found to be abnormal. Therefore, if we had also asked about whether they felt neglected by their HCP, it would have created understandable tension. We also had to restrict the number of questions we asked to prevent PEM in participants; as it was, the questionnaires were lengthy with the focus on supporting biological data.